# DOF: Accelerating High-order Differential Operators with Forward Propagation

**Ruichen Li**[1,*]  **Chuwei Wang**[2,*]  **Haotian Ye**[3,*]  **Di He**[1,†]  **Liwei Wang**[1,4,†]

[1]National Key Laboratory of General Artificial Intelligence, Peking University
[2]California Institute of Technology  [3]Stanford University
[4]Center for Machine Learning Research, Peking University

## Abstract

Solving partial differential equations (PDEs) efficiently is essential for analyzing complex physical systems. Recent advancements in leveraging deep learning for solving PDE have shown significant promise. However, machine learning methods, such as Physics-Informed Neural Networks (PINN), face challenges in handling high-order derivatives of neural network-parameterized functions. Inspired by Forward Laplacian, a recent method of accelerating Laplacian computation, we propose an efficient computational framework, Differential Operator with Forward-propagation (DOF), for calculating general second-order differential operators without losing any precision. We provide rigorous proof of the advantages of our method over existing methods, demonstrating two times improvement in efficiency and reduced memory consumption on *any* architectures. Empirical results illustrate that our method surpasses traditional automatic differentiation (AutoDiff) techniques, achieving 2x improvement on the MLP structure and nearly 20x improvement on the MLP with Jacobian sparsity.

## 1 Introduction

Partial differential equations (PDEs) play a pivotal role in understanding and predicting the behavior of physical systems. While classical numerical methods have proven effective in some cases, they can be prohibitively challenging when dealing with complicated problems, e.g., turbulence in fluid dynamics [1] and high dimensional equations [2]. Recently, the advent of deep learning[3] has spurred a wave of innovations in leveraging neural networks (NN) for numerical solutions of PDEs. These NN-based approaches have been applied to various problems, including fluid dynamics [4; 5], high-dimensional optimal control problems [6; 7], and quantum many-body problems[8; 9]. Notable works, such as the Physics-Informed Neural Network (PINN)[4; 10; 11] and Neural Operator [12; 13; 14; 15], showcase the potential of neural networks in capturing the underlying physics of systems governed by PDEs.

Among these works, the central idea is to parameterize the solution as a neural network, and optimize this expressive network with the guidance from PDE. The ever-growing *AutoDiff* packages[16; 17; 18] enables convenient calculation of associated quantities such as residual losses and derivatives, avoiding discretization errors in classical methods. However, unlike most computer vision or natural language processing tasks in which first-order derivatives are sufficient for optimizations, in PDE-relevant problems one has to deal with high-order derivatives. This raises a significant challenge as common AutoDiff packages are computationally intensive under this circumstance[19; 20]. There have been attempts to address this issue [21; 22; 23; 24]. However, they resort to either randomized methods[21; 22] or numerical differentiation[23; 24], which introduce unsatisfying statistical errors and are limited in problems where high precision is not demanded.

Recently, Li et al. [25] proposed a computational framework, Forward Laplacian (FL). This framework is designed specifically for accelerating Laplacian operator computation and thus can significantly boost the computation of Laplacian-relevant PDE like Schrödinger equation in quantum

---

*Equal contributions.
†Corresponding to: {dihe,wanglw}@pku.edu.cn.

chemistry[8]. Remarkably, Forward Laplacian is proven to be precision-preserved as it introduces no statistical errors at all. Consequently, it is natural to ask whether the idea of FL can be leveraged for other PDEs associated with high-order derivatives computation.

To fill this gap, we develop a new computational framework named Differential Operator with Forward-propagation (DOF). DOF shares a similar computational procedure as FL, yet can be applied to compute general second-order differential operators. We demonstrate that DOF outperforms conventional AutoDiff methods in memory and computational cost by a large margin, both theoretically and practically. In practice, DOF can accelerate NN-based solvers across a wide range of PDEs such as the non-homogeneous heat equation and Klein-Gordon equation.

The contribution of this paper is summarized as follows:

- We generalize the Forward Laplacian method and propose DOF to *precisely* compute arbitrary second-order differential operators of neural networks (DOF).
- We demonstrate that DOF improves computation efficiency and memory consumption simultaneously, regardless of the architecture of neural networks, both theoretically and empirically. The improvement can be significant in common architectures like MLP.

## 2 METHOD

In PINNs and many other NN-based PDE solvers, the solution of a PDE, $\phi(\mathbf{x})$, is parameterized as neural networks, $\phi(\mathbf{x}) := \phi(\mathbf{x}; \theta)$, where $\theta$ represent the NN parameters. These methods necessitate computing the high-order derivatives of a neural network. It has been shown that standard *AutoDiff* methods are not efficient for the high-order derivatives calculation[19; 20].

In this work, we focus on the calculation of the second-order differential operators, which have the form

$$\mathcal{L} : \phi(\mathbf{x}) \to \sum_{1 \le i,j \le N} a_{ij}(\mathbf{x}) \partial_{ij}^2 \phi(\mathbf{x}) + \sum_{i \le N} b_i(\mathbf{x}) \partial_i \phi(\mathbf{x}) + c(\mathbf{x}) \phi(\mathbf{x}). \tag{1}$$

Here, $a_{ij}, b_i, c : \mathbb{R}^N \to \mathbb{R}$ are coefficients in the second order operator, and $N$ is the input dimension (the time variable is comprised in $\mathbf{x}$ for evolution equations). In practice, the first term dominates the computation cost. Thus, for brevity and clarity, we will only focus on the case when $b_i \equiv c \equiv 0$ in the following discussions. We always denote the symmetric matrix $\left(a_{ij}(\mathbf{x})\right)_{i,j}$ as $A(\mathbf{x})$.

### 2.1 FORWARD LAPLACIAN

In standard AutoDiff packages, the second-order operator calculation is based on the Hessian matrix $H = (\partial_{ij} \phi(x))_{i,j}$. We call those methods *Hessian-based* methods. They use multiple Jacobian calculations to derive the Hessian matrix, resulting in a huge computation cost. Recently, Li et al. [25] proposed a new computational framework, Forward Laplacian (FL), which primarily focuses on accelerating the calculation of Laplacian, i.e., $A \equiv I_N$. Below we briefly review this method.

FL computes the Laplacian with *one* efficient forward pass, avoiding redundant calculation in the Hessian-based approach. Following the notation in Li et al. [25], we describe FL in a computation graph $\mathcal{G}$. The node set $V = \{v^i | i = 0, 1, ..., M\}$ represents the operations or variables used in a neural network. We use the abbreviation $i \to j$ if there is a directed edge from $v^i$ to $v^j$ in $\mathcal{G}$, and we denote operations as $F$, e.g., $v^j = F_j(\{v^i : i \to j\})$ for all $j \ge 0$. Notice that the node indices are arranged according to the topological order, i.e., for all $i \to j$, we have $i < j$. The output of $\phi$ is denoted by $v^M$. In addition, $\nabla$ and $\Delta$ represent the gradient operator and Laplacian operator with respect to the input, respectively. Detailed notations can be found in the appendix A.

Specifically, according to node dependency, FL sequentially computes the Laplacian tuple $(v^i, \nabla v^i, \Delta v^i)$ associated with each node. In a simplified case where $v^i$ depends only on $v^{i-1}$, i.e., $v^i = F_i(v^{i-1})$, the graph can be represented as a chain. We can compute the output tuple in a forward pass:

$$(\mathbf{x}, \nabla \mathbf{x}, \Delta \mathbf{x}) \to (v^0, \nabla v^0, \Delta v^0) \to \cdots \to (v^M, \nabla v^M, \Delta v^M) \tag{2}$$

The propagation rule of Laplacian tuple is derived through the chain rule:

$$v^i = F_i(v^{i-1}), \ \nabla v^i = \partial_{v^{i-1}} F_i \nabla v^{i-1}, \ \Delta v^i = \partial_{v^{i-1}}^2 F_i |\nabla v^{i-1}|^2 + \partial_{v^{i-1}} F_i \Delta v^{i-1} \tag{3}$$

For the general computation graph, we can generalize eq. (3) to the following formula:

$$v^j = F_j(\{v^i : i \to j\}) \tag{4}$$

$$\nabla v^j = \sum_{i:i \to j} \frac{\partial F_j}{\partial v^i} \nabla v^i \tag{5}$$

$$\Delta v^j = \sum_{\substack{i,l \\ i \to j \; l \to j}} \frac{\partial^2 F_j}{\partial v^i \partial v^l} \nabla v^i \cdot \nabla v^l + \sum_{i:i \to j} \frac{\partial F_j}{\partial v^i} \Delta v^i. \tag{6}$$

Then we can sequentially compute the Laplacian tuple for each node in the topological order according to eqs. (4) to (6). As shown in Li et al. [25], this approach outperforms the Hessian-based approach in numerous types of computational graphs. It has been successfully applied to solving the Schrodinger equation, resulting in over a magnitude of acceleration.

## 2.2 DOF FOR THE SECOND-ORDER DIFFERENTIAL OPERATORS

We now formally propose DOF to efficiently compute all kinds of second-order operators. For brevity, we denote $A(\mathbf{x})$ as $A$ and $a_{ij}(\mathbf{x})$ as $a_{ij}$ in the following discussion. To compute $\sum_{i,j} a_{ij} \partial_i \partial_j \phi(\mathbf{x})$, we first decompose the coefficient matrix $A$ into $L^\top D L$ such that $D$ is a diagonal matrix whose diagonal elements are all $\pm 1$ and 0. As $A$ is a symmetric matrix, this decomposition can be done easily. For instance, we can eigen-decompose $A = S^\top \Sigma S$, where $S$ is an orthogonal matrix and $\Sigma$ is the diagonal eigenvalue matrix, and choose $L = |\Sigma|^{1/2} S$ and $D = \text{sgn}(\Sigma)$.

During the computation, for each node $v^k$, we compute tuple $(v^j, \mathbf{g}^j, s^j) := (v^j, L\nabla v^j, \mathcal{L}v^j)$. We can derive the following formula by chain rule:

$$v^j = F_j(\{v^i : i \to j\}) \tag{7}$$

$$\mathbf{g}^j = \sum_{i:i \to j} \frac{\partial F_j}{\partial v^i} \mathbf{g}^i \tag{8}$$

$$s^j = \sum_{\substack{i,l \\ i \to j \; l \to j}} \frac{\partial^2 F_j}{\partial v^i \partial v^l} \mathbf{g}^{i\top} D \mathbf{g}^l + \sum_{i:i \to j} \frac{\partial F_j}{\partial v^i} s^i \tag{9}$$

We can derive $\mathcal{L}\phi$ by sequentially applying this propagation rule to each node in the topological order. We will prove that this method outperform the Hessian-based methods in both the memory usage and computation cost for any neural network architecture:

**Theorem 2.1.** *The computation cost (counted in FLOPs) of DOF is at most half that of Hessian-based methods for any neural network architecture.*

**Theorem 2.2.** *The memory consumption of DOF ($\mathcal{M}_1$) is smaller than that of Hessian-based methods ($\mathcal{M}_2$) for any neural network architecture.*

*Specifically, $\mathcal{M}_1 \lesssim \frac{2}{L}\mathcal{M}_2$ for an L-layer MLP.*

The proof for theorems 2.1 and 2.2 could be found in appendices B and D. We remark that the memory and computation consumption of DOF can be further reduced in some specific network architectures. See section 3.2 for details.

To better understand the DOF method, we discuss the implementation of DOF on two special classes of second-order operators.

**Elliptic Operator.** For elliptic operator, the coefficient matrix $A$ is positive so we have $D = I_N$. As a result, eqs. (7) to (9) are reduced to eqs. (4) to (6). The only difference between DOF and Forward Laplacian here is the initial value of the tuple (i.e. the tuple at node $v^j$ for $j = 0$). Thus, we utilize some existing Forward Laplacian package[26; 27] to compute the elliptic operator.

**Low-rank Coefficient Matrix.** It is well-known that computing the Hessian-vector product can be faster than computing the entire Hessian matrix in the standard AutoDiff package. Namely, computing Hessian-vector product takes $\mathcal{O}(1/N)$ cost of computing the entire Hessian matrix. Thus,

Table 1: Comparison between DOF and Hessian-based method on the MLP

| Operator | GPU Memory Usage (MB) | | | Time (ms) | | |
|---|---|---|---|---|---|---|
| | Hessian | DOF | ratio | Hessian | DOF | ratio |
| Elliptic | 10421 | 3165 | 3.3 | 196.7 | 106.6 | 1.8 |
| Low-rank | 10427 | 2141 | 4.9 | 196.2 | 55.8 | 3.5 |
| General | 10429 | 3181 | 3.3 | 197.4 | 122.2 | 1.6 |

Table 2: Comparison between DOF and Hessian-based method on the MLP with Jacobian sparsity

| Operator | GPU Memory Usage (MB) | | | Time (ms) | | |
|---|---|---|---|---|---|---|
| | Hessian | DOF | ratio | Hessian | DOF | ratio |
| Elliptic | 21401 | 997 | 21.5 | 366.4 | 18.9 | 19.4 |
| Low-rank | 21401 | 869 | 24.6 | 366.2 | 12.7 | 28.9 |
| General | 21401 | 997 | 21.5 | 366.6 | 18.9 | 19.4 |

in the standard AutoDiff package, if the coefficient matrix is a low-rank matrix, the calculation of the second-order operator can be accelerated through multiple Hessian-vector products. Similarly, the DOF method also accomplishes this acceleration when dealing with a low-rank coefficient matrix.

If the coefficient matrix $A$ is a rank-$r$ matrix, we can eliminate the columns and rows associated with the zero eigenvalues in $L$ and $D$. Then we have $L' \in \mathbb{R}^{r \times N}$ and $D' \in \mathbb{R}^{r \times r}$ that still satisfy $A = L'^\top D' L'$. Thus, while the propagation rule of DOF remains the same, the dimension of $\mathbf{g}^k$ is reduced from $N$ to $r$. According to the analysis in the appendices B and D, this reduction in the dimension will lead to an $\mathcal{O}(r/N)$ reduction in both the memory usage and the computation cost.

## 3 Results

In this section, we compare DOF with the standard Hessian-based approach on different operators and network architectures. We study three kinds of second-order operators: elliptic operator, elliptic operator with low-rank coefficient matrix, and general operator. As for the architecture, we choose the standard MLP structure and the MLP with Jacobian sparsity. Details could be found in appendix E.

### 3.1 Comparison on the MLP

The benchmark results for the MLP are shown in table 1. For both the elliptic and general operators, DOF demonstrates significant efficiency improvements, halving the computation cost and reducing memory usage by a third compared to the Hessian-based method, aligning with our theoretical predictions. In the case of the low-rank operator, DOF achieves even greater acceleration, attributed to the dimensionality reduction of $\mathbf{g}$.

### 3.2 Comparison on the MLP with Jacobian Sparsity

As discussed in Li et al. [25], the cost of Forward Laplacian method is significantly reduced when the Jacobian of intermediate component is sparse. We notice that this sparse Jacobian property also exists in some advanced PINN network architecture [28]. As a forward AutoDiff method, DOF can also leverage this property to significantly accelerate the calculation. The memory and time comparison are listed in table 2 and the architecture details can be found in appendix E. Compared with the Hessian-based method, DOF can significantly reduce both the memory and computation consumption in the MLP with Jacobian sparsity, showing a great potential in applying our method to the advanced machine learning-based PDE solver.

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

# A  PRELIMINARY

In this section, we briefly review computation framework in deep learning and auto differentiation, and introduce the notations we will use in the following sections.

## A.1  COMPUTATION GRAPH

The computation graph serves as a descriptive language of deep learning models across various deep learning toolkits, including PyTorch [17], TensorFlow [18], and Jax [16].

In the computation graph $\mathcal{G}$ associated with a function $\phi(\mathbf{x})$, $\mathbf{x} = (x_1, ...x_N) \in \mathbb{R}^N$ (we will always use $N$ for input dimension), the edges represent function arguments, and nodes represent operations or variables. Note that $\mathcal{G}$ is always a directed acyclic graph. We use $\{v^{-1}, ...v^{-N}\}$ to represent the external nodes of $\mathcal{G}$ (i.e., the input $\mathbf{x}$ of the neural network-parameterized function $\phi$), and use $\{v^0, ...v^M\}$ to represent the internal nodes, sorted in topological orders. A node can be referred to as a neuron in the network. Specifically, we use $v^M$ to serve as the network output $\phi$. We use the abbreviation $i \to j$ if there is a directed edge from $v^i$ to $v^j$ in $\mathcal{G}$. Furthermore, we denote operations as $F$, e.g., $v^j = F_j(\{v^i : i \to j\})$ for all $j \geq 0$.

**Example A.1.** *Take Multi-Layer Perceptron (MLP) function $\phi(\mathbf{x})$ as an example.*

*$\phi$ has the form $\phi = F_L \circ F_{L-1} \circ ... \circ F_0$, where $F_l = (F_{l,1}, ...F_{l,N_{l+1}}) : \mathbb{R}^{N_l} \to \mathbb{R}^{N_{l+1}}$ are the mapping in each layer with $N_0 = N$ and $N_{L+1} = 1$. Let $\mathbf{u}^l = (u_1^l, u_2^l, ...u_{N_l}^l) \in \mathbb{R}^{N_l}$ be the vector consist of the neurons in the l-th layer, we have $\mathbf{u}^0 = \mathbf{x}$ and $\mathbf{u}_{l+1} = F_l(\mathbf{u}^l) = \sigma(W^l \mathbf{u}^l + b^l)$ where $W^l \in \mathbb{R}^{N_{l+1} \times N_l}$, $b^l \in \mathbb{R}^{N_{l+1}}$ are network parameters and $\sigma$ is a nonlinear function operated element-wise.*

*In this setting, the nodes $\{v^i\}_i$ in the computation graph are*

$$x_1, ...x_N, \ u_1^1, ...u_{N_1}^1, \ u_1^{1.5}, ...u_{N_1}^{1.5}, \ u_1^2, ..., u_{N_2}^2, ..., u_1^{L+1} (= \phi(\mathbf{x})), \tag{10}$$

*and the operations generating the node representing $u_i^l$ and $u_i^{l-0.5}$ are $u_i^l = \sigma(u_i^{l-0.5})$ and $u_i^{l-0.5} = \sum_{j=1}^{N_{l-1}} W_{ij}^{l-1} u_j^{l-1} + b_i^{l-1}$, respectively.*

## A.2  AUTO DIFFERENTIATION

In most machine learning toolkits, auto differentiation(AutoDiff) implemented with back propagation algorithm is applied to compute the derivatives of neural network functions.

For a function $\phi(\mathbf{x})$, this method first performs forward propagation to obtain the value of each variable $v^j$, and construct the computation graph $\mathcal{G}$. Next, a backward process is employed by creating a new computation graph $\hat{\mathcal{G}}$ where node $\hat{v}^i \in \hat{\mathcal{G}}$ represents the operation to calculate $\frac{\partial \phi}{\partial v^i}$, $i = M, ..., -N$. The associated computations are

$$\frac{\partial \phi}{\partial v^i} = \sum_{j:i \to j} \frac{\partial F_j}{\partial v^i} \frac{\partial \phi}{\partial v^j}, \ i = M - 1, ..., -N. \tag{11}$$

# B  PROOF OF THEOREM 2.1

Our proof follows similar idea to what is discussed in section 4.2 in Li et al. [25], except that we give the proof for general second order operator here.

Recall that we are going to compute eq. (1) and we only need to analyze the computation cost of its first term. In the following, we will use the notation described in appendix A for computation graphs. The previous auto differentiation method obtains $\mathcal{L}\phi(\mathbf{x})$ through computing the Hessian matrix and its inner product with $A$. It first performs forward propagation to obtain the value of each variable $v^j$. Next, a standard backward process is employed by creating a new computation graph $\hat{\mathcal{G}}$ where node $\hat{v}^i \in \hat{\mathcal{G}}$ represents the operation to calculate $\frac{\partial \phi}{\partial v^i}$, $i = M, ..., -N$. The associated computations are

$$\frac{\partial \phi}{\partial v^i} = \sum_{j:i \to j} \frac{\partial F_j}{\partial v^i} \frac{\partial \phi}{\partial v^j}, \; i = M-1, ..., -N. \tag{12}$$

The Hessian matrix is then obtained through the following forward-mode Jacobian calculation along $\mathcal{G}$ and $\hat{\mathcal{G}}$, respectively:

$$\nabla v^i = \sum_{j:j \to i} \frac{\partial F_i}{\partial v^j} \nabla v^j, \; i = -N, ...M \tag{13}$$

$$\nabla \hat{v}^i = \nabla \frac{\partial \phi}{\partial v^i} = \sum_{\substack{j,l \\ i \to j \\ l \to j}} \frac{\partial^2 F_j}{\partial v^l \partial v^i} \frac{\partial \phi}{\partial v^j} \nabla v^l + \sum_{j:i \to j} \frac{\partial F_j}{\partial v^i} \nabla \frac{\partial \phi}{\partial v^j}, \; i = M-1, ... -N \tag{14}$$

where we always use $\nabla$ to denote $\nabla_\mathbf{x}$ for simplicity.

The bottleneck of Hessian computation comes from eq. (13) and eq. (14).

For every connected nodes $(j \to i)$, it takes $N$ float multiplication to yield $\frac{\partial F_i}{\partial v^j} \nabla v^j$. Therefore eq. (13) takes $N|E|$ floating point operations (FLOPs) if we only count multiplications, where $E$ denotes the set of edges in $\mathcal{G}$. For eq. (14), the second term also takes $N|E|$ FLOPs. To calculate the computational cost of the first term in eq. (14), we first introduce two notations, $T$ and $R$, which are both sets of ordered tuples:

$$T = \{(i,l,j)|i \to j, \; l \to j, \; \frac{\partial^2 F_j}{\partial v^i \partial v^l} \neq 0\},$$
$$R = \{(i,l)|\exists j \; s.t. \; (i,l,j) \in T\}. \tag{15}$$

The AutoDiff method sums over $j$ first to obtain $\sum_{j:i \to j, l \to j} \frac{\partial^2 F_j}{\partial v^l \partial v^i} \frac{\partial \phi}{\partial v^j}$ for all $(i,l) \in R$, and then sums over $l$. For the first step, by leveraging the symmetry of Hessian matrix, it spends 1 FLOPS for a pair of $(i,l,j)$ and $(l,i,j)$ in $T$. Thus it spends $0.5|T|$ FLOPs in total. For the second step, for any $(i,l) \in R$, it takes $N$ FLOPs to multiply a vector $\nabla v^l$ with a scalar. Thus this steps takes $N|R|$ FLOPs in total. Consequently, the total FLOPs for the previous method is about $N(|R| + 2|E|) + 0.5|T|$.

Next we analyze the computation cost of DOF. For readers' convenience, we repeat the propagation scheme here:

During the computation, for each node $v^k$, we maintain a tuple $(v^k, \mathbf{g}^k, s^k) := (v^k, L\nabla v^k, \mathcal{L}v^k)$. The propagation rule of this tuple is:

$$v^j = F_j(\{v^i : i \to j\}) \tag{16}$$

$$\mathbf{g}^j = \sum_{i:i \to j} \frac{\partial F_j}{\partial v^i} \mathbf{g}^i \tag{17}$$

$$s^j = \sum_{\substack{i,l \\ i \to j \; l \to j}} \frac{\partial^2 F_j}{\partial v^i \partial v^l} \mathbf{g}^{i\top} D \mathbf{g}^l + \sum_{i:i \to j} \frac{\partial F_j}{\partial v^i} s^i \tag{18}$$

For the proposed DOF method, we perform forward propagation along $\mathcal{G}$ to obtain $\phi(\mathbf{x})$, $L\nabla\phi(\mathbf{x})$ (recall that $L$ is a matrix that comes from the decomposition of $A$) and $\mathcal{L}\phi(\mathbf{x})$. The second term

$L\nabla\phi(\mathbf{x})$ is calculated along $\mathcal{G}$ according to eq. (17). The third term, i.e., the target differential operator, is calculated along $\mathcal{G}$ according to eq. (18).

The computational cost of the DOF method is dominated by eq. (17) and eq. (18). As previously discussed, eq. (17) takes $N|E|$ FLOPs. For the first term in eq. (18), we decompose its calculation into two steps. First, we compute $\{\mathbf{g}^{i\top}D\mathbf{g}^l\}_{i\leq l,\ (i,l)\in R}$. Since $D$ is a diagonal matrix only with diagonal element in $\{0,\pm1\}$, this computation takes $0.5\mathrm{rank}(D)|R| \leq 0.5N|R|$ FLOPs in total. Next, following the topological order of $\mathcal{G}$, we sum over $i \leq l$ for each $j$, deriving the first term in eq. (18). By leveraging the symmetry of the Hessian matrix and Gram matrix, we reduce this computation by a factor of 2, which is $0.5|T|$ FLOPs. The computational cost of the second term is negligible compared with the first term since $s^i$ is a scalar.

Summing the computational cost of all the terms, we have that the DOF method uses at most $0.5N(|R| + 2|E|) + 0.5|T|$ FLOPs.

In practice, a large percentage of operations are linear transformations, and for any linear operation $F_j$, $\frac{\partial^2 F_j}{\partial v^i \partial v^l} = 0$ for any $i \to j$, $l \to j$. This means the value $|T|$ is much smaller than $N|R|$ and $N|E|$. Thus, our method is about two times faster than the previous *Hessian-based* methods for computing second order differential operators of general neural network functions.

## C  CASE STUDY FOR MLP

In this section, we follow the notation in appendix A for an MLP $\phi$ and show that there is further speedup comparing with the general 2x result stated in theorem 2.1.

For MLP, We could explicitly compute

$$|E| = \sum_{l=0}^{L} N_l N_{l+1}, \ |T| \leq \sum_{l=0}^{L} N_{l+1} N_l (N_l - 1), \ |R| = \sum_{l=0}^{L} N_l (N_l - 1). \tag{19}$$

To compute the first term in eq. (18), note that

$$\frac{\partial F_{k,i}}{\partial u_j^k} = \sigma'(W^k \mathbf{u}^k + b^k)_i W_{ij}^k \tag{20}$$

$$\frac{\partial^2 F_{k,i}}{\partial u_j^k \partial u_l^k} = \sigma''(W^k \mathbf{u}^k + b^k)_i W_{ij}^k W_{i,l}^k, \tag{21}$$

so we actually have

$$\sum_{j,l} \frac{\partial^2 F_{k,i}}{\partial u_j^k \partial u_l^k} (L \nabla u_j^k)^\top D(L \nabla u_l^k) \tag{22}$$

$$= (\frac{\sigma''}{\sigma'^2}(W^k \mathbf{u}^k + b^k))_i (L \nabla u_i^{k+1})^\top D(L \nabla u_i^{k+1}). \tag{23}$$

This gives an alternative way to compute the first term in (18), whose total computation cost is reduced from $r(D)|R|$ to $r(D) \sum_{l=0}^{L} N_{l+1}$ FLOPs, where r stands for $rank$.

Recall that the first term of (18) is one of the dominant calculations, this manner certainly boost the efficiency.

## D  PROOF OF THEOREM 2.2

We focus on the peak memory usage caused by the forward-mode Jacobian calculation, i.e., the storage of $\nabla v^i$(or $\mathbf{g}^i$), which usually dominates the memory cost. In a typical forward-mode Jacobian calculation, we write $\nabla v^i$ into memory when the algorithm is applied to $v_i$ and remove it from memory when all the direct subnodes of $v^i$ have been computed.

We denote

$$\tau(i) := \max\{j : i \to j\}. \tag{24}$$

Then at the moment our DOF forward propagation comes to the node $v^j$, the memory consumption is

$$C(j) := N \sum_{i:i \leq j \leq \tau(i)} 1. \tag{25}$$

If we further denote $\mathcal{M}_1$ to be the peak memory usage when executing the forward-mode DOF, we have

$$\mathcal{M}_1 = \max_j C(j). \tag{26}$$

From eq. (25) we clearly see that $\mathcal{M}_1 \leq N|V|$, here $V$ is the set of nodes in the computation graph $\mathcal{G}$. Furthermore, for any $j < M$, $C(j) \leq N(|V| - 1)$. As a result, $\mathcal{M}_1 = N|V|$ if and only if every nodes is pointing to the end node $v^{\overline{M}}$, which corresponds to either one-layer linear model or a multivariate elementary function taking in input $\mathbf{x}$ and directly giving the output. These extreme neural network functions rarely occur in deep learning literature.

By contrast, in the Hessian calculation in AutoDiff, the computation graph becomes a combination of $\mathcal{G}$ and $\hat{\mathcal{G}}$. As suggested by eq. (14), the node $\hat{v}^i$ is a direct subnode of $v^i$. Consequently, at the moment when $\nabla v^M$ is written into memory, every $\nabla v^i$ for $v^i \in \mathcal{G}$ have been written into memory and could not be released since $\hat{v}^i$ have not been computed yet. As a result, the peak memory usage(denoted as $\mathcal{M}_2$) is strictly larger than $N|V|$, and therefore larger than the peak memory of Forward Laplacian.

In the specific case when $\phi$ is MLP. For any node $u_i^k$, $k \in \frac{1}{2}\mathbb{N}$,

$$\tau(\#u_i^k) = \begin{cases} \#u_{N_{k+1}}^{k+0.5}, & k = l \\ \#u_i^{k+0.5}, & k = l + 0.5, \ l \in \mathbb{N}, \end{cases} \tag{27}$$

where $\#$ denotes the order index of the node representing certain neurons in $\mathcal{G}$. Thus,

$$\frac{1}{N}C(\#u_i^k) = \begin{cases} N_l + 1, & k = l \\ N_{l-1} + i, & k = l + 0.5, \ l \in \mathbb{N}. \end{cases} \tag{28}$$

Thus we conclude that $\mathcal{M}_1 \leq N \max_l N_l + N_{l-1} \lesssim N\frac{2}{L}\sum_l N_l = N\frac{2}{L}|V| \leq \frac{2}{L}\mathcal{M}_2$. $\qquad\square$

# E  EXPERIMENTS SETTINGS

**Hardware.**   All the results are evaluated on a single NVIDIA Tesla V100 GPU.

**Network structure.**   We use the MLP and MLP with Jacobian sparsity to benchmark different methods. The MLP with Jacobian sparsity means we split each data into some small blocks $\mathbf{x} = (\mathbf{x}_1, \mathbf{x}_2, ... \mathbf{x}_k)$ and independently operate each block with an MLP. The output is a sum of product of each MLP output, i.e.

$$\text{output} = \sum_d \prod_{i=1}^{k} [\text{MLP}^i(\mathbf{x}_i)]_d.$$

Here $i$ refers to the index of block and $d$ refers to the index of the output in each MLP. Detailed hyperparameters can be found in table 3.

Table 3: Hyperparameters

|                               | MLP | MLP with Jacobian sparsity |
|-------------------------------|-----|----------------------------|
| hidden dimension              | 256 | 256                        |
| input dimension               | 64  | 64                         |
| #layer                        | 8   | 8                          |
| #blocks                       | -   | 16                         |
| output dimension for each MLP | -   | 8                          |

**Coefficient matrix used for each experiments.**   For the MLP structure, the coefficient matrix $A$ is a $64 \times 64$ matrix. For the MLP with Jacbobian sparsity, the coefficient matrix $A$ is a $64 \times 64$ block diagonal matrix. The coefficient matrices are listed in table 4. Here, $(\alpha_{ij})_{i,j}$ is a $64 \times 64$ matrix and $(\sigma_{ij})_{i,j}$ is a $4 \times 4$ matrix. Both $\alpha_{ij}$ and $\sigma_{ij}$ are drawn from a standard normal distribution. $(\delta_{ij})_{i,j}$ is the identity matrix such that $\delta_{ij} = 0$ if $i \neq j$ else 1. $s_i = -1$ if $i = 0$ else 1. .

Table 4: Coefficient matrix

| Structure         | Elliptic                                          | Low-rank                                          | general                                |
|-------------------|---------------------------------------------------|---------------------------------------------------|----------------------------------------|
| MLP               | $a_{ij} = \sum_{k=1}^{64} \alpha_{ik}\alpha_{jk}$ | $a_{ij} = \sum_{k=1}^{32} \alpha_{ik}\alpha_{jk}$ | $a_{ij} = \delta_{ij}s_i$              |
| MLP with sparsity | $a_{il,jm} = \delta_{lm}\sum_{k=1}^{4} \sigma_{ik}\sigma_{jk}$ | $a_{il,jm} = \delta_{lm}\sum_{k=1}^{2} \sigma_{ik}\sigma_{jk}$ | $a_{il,jm} = \delta_{lm}\delta_{ij}s_i$ |

