# OpenReview forum: "DOF: Accelerating High-order Differential Operators with Forward Propagation"
_ICLR.cc/2024/Workshop/AI4DiffEqtnsInSci — AI4DiffEqtnsInSci @ ICLR 2024 Poster_

### Official Review · Reviewer_dkS4 · 2024-02-22
**Technically solid idea with provable efficiency claims**

**Rating:** 7
**Confidence:** 3

**Review:**

**Summary:** The paper proposes a new way to compute second-order differential operators using automatic differentiation. Authors extend on the recent results of the Forward Laplacian method by Li et al. to add arbitrary second-order differential operators with provably faster and lower memory calculations. The paper showcases results on MLP with/without Jacobian sparsity.

**Strengths:**
1. The paper reads well and is easy to follow. I enjoyed reading the paper.
2. The authors address a significant concern on the efficient calculation of second-order differential operators.
3. Convicing analysis of performance in domains of both time and memory.

**Areas of Improvement:**
1. The paper does not mention efficient higher-order derivative calculations like Taylor mode AD [bettencourt2019, tan2023]. The works relate greatly, and I would like to see performance comparisons.
2. How well does this technique generalize to compute arbitrary higher-order derivatives?
3. Open-source software: Do the authors plan to present their algorithm as open-source software?

[bettencourt2019] Bettencourt, Jesse, Matthew J. Johnson, and David Duvenaud. "Taylor-mode automatic differentiation for higher-order derivatives in JAX." Program Transformations for ML Workshop at NeurIPS 2019. 2019.

[tan2023] Tan, Songchen. Higher-Order Automatic Differentiation and Its Applications. Diss. Massachusetts Institute of Technology, 2023.

---

### Official Review · Reviewer_nEYx · 2024-02-25
**This submission aims to propose the Differential Operator with Forward-propagation (DOF) for solving partial differential equations and more specifically for calculating general second-order differential operators without losing any precision. The authors’ empirical results illustrate that their method surpasses traditional automatic differentiation (AutoDiff) techniques, achieving 2x improvement on the MLP structure and nearly 20x improvement on the MLP with Jacobian sparsity.**

**Rating:** 9
**Confidence:** 5

**Review:**

A) One major limitation of this work is that it only refers to physics inspired neural networks and ignores much that has been done in the last 5 years such as MGNO, Fourier neural operators or the multiwavelet neural operators. Here are some recent examples that solve similar problems:
- “Multipole graph neural operator for parametric partial differential equations” In Advances in Neural Information Processing Systems, volume 33, pages 6755–6766, 2020.
- "Non-linear operator approximations for initial value problems." In International Conference on Learning Representations (ICLR). 2022.
- "Coupled Multiwavelet Operator Learning for Coupled Differential Equations." In The Eleventh International Conference on Learning Representations. 2022.
- "Multiwavelet-based operator learning for differential equations." Advances in neural information processing systems 34 (2021): 24048-24062.
- "Identifying arguments of space-time fractional diffusion: data-driven approach." Frontiers in Applied Mathematics and Statistics (2020): 14.
- "Fourier neural operator with learned deformations for pdes on general geometries." arXiv preprint arXiv:2207.05209 (2022).
B) The authors stated in the article that the use of DOF will not cause a loss in precision. Can this be specifically shown in the experimental results section?
C) The authors compared the proposed strategy with the AutoDiff method based on the Hessian matrix. Can more methods be added to prove the improvement of DOF?
D) The authors need to improve the rigor of the paper’ writing. Many of the citations that appear in the article are in the wrong order. Also as highlighted above they ignored major developments in this field of many papers that collected hundreds of citations, so I see no reason for which they ignore good prior work.

---

### Meta-Review · Area_Chair_gYye · 2024-02-25

**Recommendation:** Accept (Poster)

**Metareview:**

Both reviewers mark this paper as a strong accept. I recommend adding the references suggested by Reviewer nEYx and I vote for acceptance.

---

### Decision · Program_Chairs · 2024-02-28

Accept (Poster)